



# Influence of Storm Type on Compound Flood Hazard of a Mid-Latitude Coastal-Urban Environment

Ziyu Chen[1*], Philip M. Orton[1], James Booth[2], Thomas Wahl[3], Arthur DeGaetano[4], Joel Kaatz[5], Radley M. Horton[6]

[1]Department of Civil, Environmental and Ocean Engineering, Stevens Institute of Technology, Hoboken, 07030, USA
[2]Department of Earth and Atmospheric Sciences, City University of New York, City College and Department of Earth and Environmental Sciences, City University of New York, The Graduate Centre, New York, 10016, USA
[3]Department of Civil, Environmental and Construction Engineering, and National Centre for Integrated Coastal Research, University of Central Florida, Orlando, 32816, USA
[4]Northeast Regional Climate Centre Department of Earth and Atmospheric Sciences, Cornell University, Ithaca, 14853, USA
[5]Arcadis, New York, 10279, USA
[6]Columbia Climate School, Columbia University, 10025, USA

*Correspondence to*: Ziyu Chen (zchen44@stevens.edu)

**Abstract.** A common feature within coastal cities is small, urbanized watersheds where the time of concentration is short, leading to vulnerability to flash flooding during coastal storms that can also cause storm surge. While many recent studies have provided evidence of dependency in these two flood drivers for many coastal areas worldwide, few studies have investigated their co-occurrence locally in detail, nor the storm types that are involved. Here we present a bivariate statistical analysis framework with historical rainfall and storm surge and tropical cyclone (TC) and extratropical cyclone (ETC) track data, using New York City (NYC) as a midlatitude demonstration site where these storm types play different roles. In contrast to prior studies that focused on daily or longer durations of rain, we apply hourly data and study simultaneous drivers and lags between them. We quantify characteristics of compound flood drivers including their dependency, magnitude, lag time and joint return periods, separately for TCs, ETCs, non-cyclone associated events, and merged data from all events. We find TCs have markedly different driver characteristics from other storm types and dominate the joint probabilities of the most extreme rain-surge compound events, even though they occur much less frequently. ETCs are the predominant source of more frequent, moderate compound events. The hourly data also reveal subtle but important spatial differences in lag times between the joint flood drivers. For Manhattan and southern shores of NYC during top-ranked TC rain events, rain intensity has a strong negative correlation with lag time to peak surge, promoting pluvial-coastal compound flooding. However, for the Bronx River in northern NYC, fluvial-coastal compounding is favoured due to a 2-6 hour lag from the time of peak rain to peak surge.



## 1 Introduction

Floods are one of the most catastrophic natural hazards, and frequently threaten human life and property worldwide. They are normally classified by several types such as coastal, pluvial and fluvial flooding based on their triggering mechanisms, also known as flood drivers (Pachauri and Reisinger, 2007). These flood drivers can occur at the same time to cause what is referred
to as compound flooding (e.g. Field et al., 2012).

Flood hazard assessments traditionally focus on evaluating the extreme values of coastal and riverine or rainfall drivers separately (e.g. Arns et al., 2013; Abdelkader et al., 2023; Perica et al., 2013; Ayyad et al., 2023). However, this may underestimate hazard risk by neglecting co-occurrence of two or more extremes in a single event (Zscheischler et al., 2018). Beginning after Hurricane Katrina (2005), the joint occurrence of multiple extremes is emphasized and documented in the new
flood protection construction criteria (Resio et al., 2007).

In response to this growing understanding of the risk from compound flood hazards, the topic has been studied in several ways, including statistical analyses of correlations and joint probabilities of the historical extremes of the compound flood drivers (e.g. Wahl et al., 2010) hydrologic and hydrodynamic modelling of historical events (e.g. Orton et al., 2012; Mita et al., 2023) or hybrid approaches using synthetic storms (e.g. Gori et al., 2022).

In statistical analyses of historical compound extreme flood drivers, multivariate statistical models (e.g. Najibi et al., 2023; Jane et al., 2020; Wahl et al., 2012) and frameworks (e.g. Bender et al., 2016; Torre et al., 2019) are developed to improve modelling of the dependency structure of the compound correlated extremes. Copula theory (Sklar, 1959) is commonly applied as a solution for multivariate probabilistic modelling, since Copulas (e.g. Joe, 2014; Roch and Alegre, 2006) have flexible joint distributions to quantify the dependency of correlated events. Bivariate or trivariate combinations of flooding factors such
as waves, storm surge, water level, river discharge and volume, rainfall intensity and duration, groundwater, sea level rise, etc., are selected for the multivariate statistical analysis based on various research interests (e.g. Kim et al., 2023; Lai et al., 2021; Moftakhari et al., 2017; Sadegh et al., 2017; Salvadori et al., 2014; Ward et al., 2018; Al Azad et al., 2018). Those studies have been done at global (e.g. Ward et al., 2013), national (e.g. Wahl et al., 2015) regional (e.g. Gori et al., 2020b) and local scales (e.g. Jane et al., 2020).

Most of these studies quantify the compound flood events from a single population dataset without distinguishing what meteorological systems are causing them. Tropical cyclones (TCs, including post-tropical cyclones) and extratropical cyclones (ETCs) both can cause coastal hazard extremes and compound flooding. However, these storm types have different energy and moisture sources and cause different hazard intensities in terms of maximum wind speed, storm surge (e.g. Orton et al., 2016; Ayyad et al., 2022; Chen et al., 2019) and rainfall rates. ETCs normally have a larger spatial extent and have wind speeds
far below the maxima exhibited by TCs (e.g. Dolan and Davis, 1992; Landsea and Franklin, 2013) and TCs can have more abundant moisture. Each storm type has often been shown to exhibit different univariate extreme value probability distributions (Lin et al., 2010; Villarini and Vecchi, 2013; Orton et al., 2016). Each storm type thus may also have distinct compound flood hazard characteristics, pointing to the importance of not assessing all storm events together as one population (Orton et al.,





2016). However, this separation by storm type has only rarely been attempted in past studies of compound flooding (e.g. Kim
et al., 2023), especially in mid-latitude areas that are affected by both TCs and ETCs.

Also, many studies ignore the question of relative timing of the drivers within a storm, using the storm-maximum flood drivers
rather than the simultaneous ones as the compound sampling pairs. It is unclear in these studies whether the flood drivers are
compounding or sequentially occurring. Due to the more widespread and longer duration historical archives of daily versus
hourly rain data, most prior studies have used daily data. This limits our understanding of relative timing of drivers and is not
an appropriate timescale to understand pluvial flooding processes.

The urban environment has a much larger proportion of unvegetated impervious surfaces and additional vulnerability due to
dense population and extensive infrastructure. There is little infiltration of water into soils, and stormwater systems are often
insufficient to convey heavy rainfall, leading to flooding from the backing up of water (e.g. Villarini et al., 2009). Short
duration (hourly or sub-hourly) intensified rain values have been found to be the predominant driver of pluvial flooding for
NYC (New York City, 2021) and other urban environments (Rosenzweig et al., 2018). Its timing relative to the peak of coastal
water levels is a critical factor for compound flooding (Gori et al., 2020a; Xu et al., 2023). While pluvial flood research has
often utilized hourly and sub-hourly data, much of the past research on compound flooding mentioned above has relied upon
daily rainfall data.

In this study, we address the above weaknesses and demonstrate a framework for assessing compound rain-surge hazard for
different storm types and applying hourly data, using NYC as a demonstration site. We evaluate the compound hazard
characteristics from separate populations of TC and ETC events, as well as events that are attributed to "Neither" type of storm
and "All" events combined. Our framework is tailored to the compound flood risk of a typical urban pluvial flood environment
where the peak flood depth occurs relatively rapidly after the peak rainfall (i.e. the time of concentration is below one hour).
We use higher-resolution spatiotemporal data to study compound flood driver characteristics that have not been sufficiently
evaluated in prior national and global studies.

Below, Section 2 introduces the study area and data for this research; Section 3 gives a full picture of methodology from pre-
processing the data, identifying compound events, storm type association analysis methods for multiple aspects of compound
characteristics. Section 4 shows the results under the above-described framework. Section 5 discusses the key results,
limitations, and future work for this research, and Section 6 summarizes our study's conclusions.

## 2 Study Site and data

### 2.1 Study Site

NYC is the most densely populated city in the United States with more than eight million residents, 778 km² of area, and
approximately 70% impervious land coverage. It is vulnerable to pluvial and coastal flooding, and likely compound flooding
from both (e.g. Sarhadi et al., 2024; Georgas et al., 2014; Chen and Orton, 2023). It consists of several small, urbanized
watersheds (ranging from 4.7-60 km²), where the time of concentration is short. The Bronx River is one exception, a small



river with an elongated 39 km long, 105 km² watershed and by far the largest inland stream passing through NYC. The city is located in a low-elevation region with a lengthy coastline subject to flooding from both the NY/NJ Bight to the south and Long Island Sound to the northeast. It has extensive coastal floodplains along its adjacent tidal water bodies (Fig. 1). A recent study found that many of the NYC neighbourhoods with the most flood complaints are in these coastal areas (Agonafir et al., 2022);

compound flooding could be an important contributing factor. Another study used 311 flood report phone call data and found relationships between flood reports and other spatial data sets (Smith and Rodriguez, 2017).

Historically, severe coastal floods (e.g. Hurricane Sandy in 2012, a Nor'easter in December 1992), pluvial floods (e.g. Hurricane Ida in 2021) and compound floods (Hurricane Irene in 2012; Orton et al., 2012) struck NYC, and can be associated with TCs, ETCs and convective thunderstorms. Four of NYC's top-5 storm surges from 1788-present were TC (or post-tropical

cyclones), 3 of the top-5 hourly rain events from 1948-present were TC (KNYC: Central Park), and 4 of the top-5 daily rain events from 1869-present were TC (KNYC: Central Park).

## 2.2 Historical observations

In the interest of using a long-term database to study compound flooding, rain and coastal water level data were assembled with the longest possible hourly data resolution. Given that there is hourly tide gauge data back to the 1800s (Talke et al.,

2014), the limitation on data availability came from hourly rain gauge data, which were continuously available for several NYC region gauges from 1948 to present.

### 2.2.1 Tide gauges

Hourly water level data are obtained from NOAA tide gauges in UTC time zone and North American Vertical Datum of 1988 (NAVD88), including gauges (blue points in Fig. 1) at the Battery (8518750), Kings Point (8516945, from 1999 to 2022) and

Willets Point (8516990, from 1948 to1998). The locations of Kings Point and Willets Point are only 3 km apart along the East River and have similar storm surge (O'donnell and O'donnell, 2012). We merge their data to represent storm surge conditions for Northern Queens and South Bronx, simply referring to the joint dataset as Kings Point.

### 2.2.2 Rain gauges

Observed hourly rainfall data at and around NYC (red points in Fig. 1) were obtained from the NOAA National Climatic Data

Centre (NCDC). The hourly rainfall data can be used to capture the short duration (hours) rain intensity, which is critical to the pluvial flood impacts for an urban environment like NYC. Also, the hourly temporal resolution can be simultaneously matched with the hourly data around its nearby tide gauge to study simultaneous or lagged occurrences. The rain gauges we selected are all within 20 km of the coast and at elevations below 100 m and have near complete long-term temporal coverage from 1948 to 2022. Each of them is within 30 km of the Battery or Kings Point tide gauge. Overall, these gauges have good

spatial coverage around the NYC area.





### 2.2.3 TC and ETC tracks

Datasets of 6-hourly storm location and time (cyclone tracks) are utilized to associate the rainfall/coastal flood drivers with specific storm types (see Section 3.4). The TC tracks dataset is from the National Hurricane Centre (HURDAT2; Landsea and Franklin, 2013). The ETC tracks are obtained by running an automated cyclone tracking algorithm on the ERA5 reanalysis.

The tracking algorithm used is the MAP Climatology of Midlatitude Storminess (MCMS) documented in Bauer et al. (2016). MCMS identifies closed low pressure locations in the sea level pressure field and then links the low centres through time. The algorithm was developed explicitly for tracking ETCs, however, MCMS sometimes identifies TCs. Before using MCMS, the TCs tracks are removed by identifying matching tracks in HURDAT2 and the MCMS output.

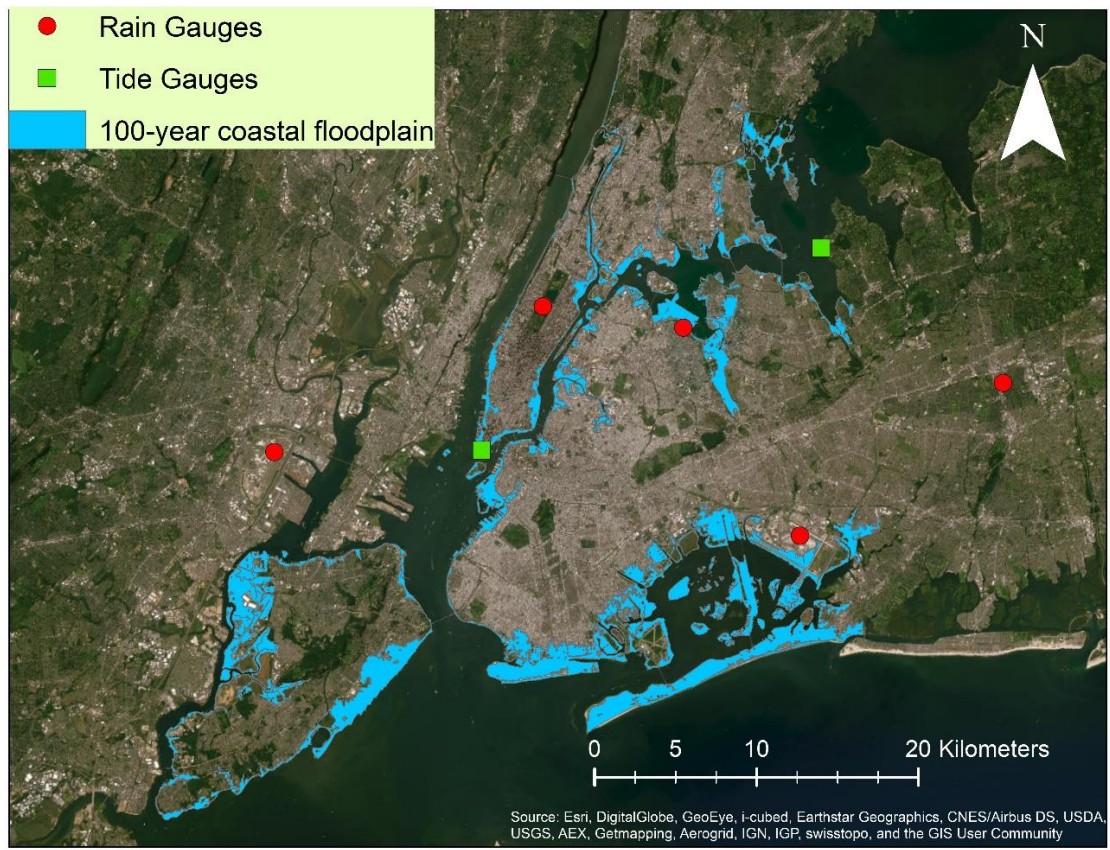

**Figure 1: Map of the locations of rain gauges (red) and tide gauges (green) around NYC. The low-elevation 100-year coastal floodplain (Fema, 2014) is shaded blue, where pluvial and coastal compound floods are more likely to occur during storms than more elevated areas (see Section 3.3). Listed from left to right, the rain gauges are Newark Liberty Airport (EWR), Central Park, LaGuardia Airport (LGA), John F. Kennedy Airport (JFK), and Mineola, and the tide gauges are Battery and Kings Point.**



## 3 Methods

### 3.1 Extreme rainfall

### 3.1.1 Rain gauges and metro-scale rain

We compute the spatial average rain within 30 km of each tide gauge separately (the Battery and Kings Point in Fig. 1) to represent the metro-scale rainfall, while each single rain gauge represents rainfall in each smaller local sewershed. Each individual rain gauge captures extreme rainfall intensity of local convective rain events and the spatial variability during
localized convective and synoptic events (TCs or ETCs). For example, TC Ida 2022 caused extreme rainfall at Central Park and LGA airport, but much less rainfall at JFK airport.

### 3.1.2 Top-ranked rain intensity of different durations

We temporally accumulate the continuous hourly rainfall data to different durations (from 1 to 48 hours) on both single rain gauges and metro-scale averaged rainfall and isolate the extreme values of the accumulated rainfall. To guarantee event
independence we eliminate peaks that occur within 5-day windows.

### 3.2 Top-ranked storm surge or non-tidal residual

We compute and remove the annual mean sea level (AMSL) from the water level data of each tide gauge. Then, we perform harmonic analysis (Schureman, 1994) on the resulting data year by year to obtain the tidal signals across the time series. Thirty-seven harmonic constituents are considered including the solar annual constituent and solar semi-annual constituent. Then, we
compute the non-tidal residual (NTR) by removing the AMSL and tide from the total water levels, thus removing SLR from the NTR data. The NTR is mainly composed of storm surge (driven by wind and atmospheric pressure), but also includes smaller contributions from river runoff and rainfall. It excludes the sea level rise (SLR) and interannual and seasonal variabilities. Prior hydrodynamic model experiments for Tropical Storm Irene (2011) showed that the effect of local rainfall on harbour water levels was only 2 cm at the time of peak water levels (Orton et al., 2012). Similar to rain data processing in
Section 3.1.2, we capture the peak 1-hour NTR and ensure event independence with a 5-day window.

### 3.3 Compound events

Storm surge and rainfall have several important differences that motivate different treatment in the sampling of compound events. Storm surge can be positive and negative, whereas the minimum rainfall is zero. Surge tends to be relatively stable over several hours' duration, while rainfall can be extreme for one hour, zero for the next (e.g. for a convective storm or as
spiral rain bands of hurricane pass a location; Senn and Hiser, 1959). However, surge often peaks with a large positive value as a storm passes, then drops to a negative value within hours due to reversed, offshore winds (e.g. for a passing tropical cyclone; Ayyad et al., 2022). As a result, we use surge maxima, to avoid averaging negative and positive values within a





storm's passage. Our statistical analyses are conditioned on the primary flood driver being top-ranked, while the secondary flood driver can be of any value. Thus, we study events conditioned on top-ranked rain (peak 1 to 48 hours accumulations),

which we refer to as Pluvial-Coastal (P-C) events, and top-ranked NTR (peak 1-hour intensity), which we refer to as Coastal-Pluvial (C-P) events. The primary flood drivers are sampled by peaks-over-threshold (POT) approach based on a per-year average (PYA) frequency of five events. This two-sided conditional sampling approach with a POT of certain percentile (e.g. 0.95 or 0.99) is typically used in many studies to identify compound extreme events from two conditioned aspects (Jane et al., 2020; Wahl et al., 2015; Ward et al., 2018).

The relative timing of the compound flood drivers is critical to their compound effects, especially for an urban environment. Here, we investigate the characteristics of the simultaneous hourly flood drivers (Section 4.2.2, 4.3, 4.5). For comparison to common practice (e.g. Wahl et al., 2015; Lai et al., 2021; Lai et al., 2023), we also evaluate the characteristics of storm-duration (non-simultaneous) maximum flood drivers (Section 4.2.1), defined as having the secondary flood driver occur within ±1.5 day of the peak of the primary flood driver. We also study the lag times between these storm-duration maximum flood

drivers. Considering the joint occurrence of rain and surge, when the peak rain intensity occurs several hours away from the peak storm surge, these two may result in more of a sequential pair of flood events with little exacerbation from compounding. However, if they co-occur at or near the same hour, the resulting compound flood magnitude may be substantially increased (e.g. Gori et al., 2022). For Pluvial-Coastal compound flood events, the compounding typically arises due to the drainage of the urban stormwater system being blocked by the simultaneous coastal high water levels (e.g. Gold et al., 2022). For Coastal-

Pluvial compound flooding, a coastal flood can be aggravated by simultaneous rainfall (e.g. Orton et al., 2012).

### 3.4 Storm type association

Cyclone tracks (Section 2.2.3) are used to associate the top-ranked rainfall and NTR events with specific TC and ETC events or to determine if they are a "Neither" case that doesn't match any cyclone track. If the top ranked rain/NTR events occur within 500 km of the centre of a TC or within 1000 km of the centre of an ETC, the event is considered as association with TC

or ETC. TC events include those that may have become post-tropical because these storms often continue to have unusually high winds and moisture. We additionally evaluate all the events together ("All", see the diagram in Fig. 2).
Similar distances are used to judge the storm association in other studies (e.g. Kim et al., 2023; Lai et al., 2021). In the supplementary material (SM), we test the sensitivity of additional distances for storm association.





Figure 2: Diagram for the workflow.

## 3.5 Dependence analysis of compound flood drivers

Kendall's rank correlation coefficient (Kendall, 1938) between flood drivers is computed for both the "storm maximum" and the "simultaneous" cases for each storm type to assess their dependency. In addition, the non-parametric upper tail dependence coefficients (UTDCs) (Schmidt and Stadtmüller, 2006; Wahl et al., 2015) are used to check the dependence of values in the upper tail region as a consistency check with their overall rank correlation. The UTDC represents the probability of a second driver being in the upper tail region, conditional on the primary driver being in the upper tail.



### 3.6 Lag time of storm-duration maximum flood drivers

The lag time between the maximum flood drivers reflects another aspect of compound flood risk (e.g. Jane et al., 2020). We identify the timing of the maximum flood drivers of each compound event and define the "lag time" as $T_{peak\ surge}$ - $T_{peak\ rain}$, so
a positive lag means the rain peak occurs before the NTR peak. Here, our main purpose is to compare the lag time characteristics of compound events associated with different storm types, as well as to contrast the difference between the Battery and Kings Point. We also use the Kendall's rank correlation coefficient to evaluate the dependence between the primary and secondary flood driver and the absolute lag time.

### 3.7 Magnitude of compound flood drivers

Prior research often focused only on rank correlations and copula modelling of joint probabilities for assessing compounding. While a high rank correlation reflects tight coupling between drivers, it is not a prerequisite for extreme compound hazard. If the secondary driver is often extreme but does not have a high rank correlation with the primary driver, there can still occasionally be co-occurrence of both drivers' extremes.

To capture the high-end intensity of the secondary flood drivers and provide an alternative method for understanding the
potential magnitude of compounding that comes from different storm types, we compute the 50th and 90th percentiles (empirical quantiles) of the rain and NTR for the hours around the time of peak rain (for P-C compound event) and hours around the time of peak NTR (for C-P compound event). This analysis compares the "simultaneous" compound hourly intensity at a range of times around the peak (±10 hours), for a consistency check with lag time characteristics analysed in Section 3.6. This approach is similar to looking at fitted marginal distributions but is an empirical approach, without any
scaling by annual frequency. It also has an added benefit of being more straightforward for risk communication than rank correlations and copula models.

### 3.8 Magnitude of compound flood drivers

The joint probabilities and return periods of rainfall and NTR are resolved by using a recently developed copula software for bivariate analysis of compound hazards by Sadegh et al. (2018), known as Multi-hazard Scenario Analysis Toolbox (MhAST).
This toolbox is utilized to assess 17 marginal distributions (e.g. GPD) with quantile-quantile plots and Chi-Square tests and identify the optimal one based on the Bayesian Information Criterion (BIC) (Sadegh et al., 2017). Similarly, the bivariate dependency structure is analysed by fitting 25 copula models (e.g. Nelsen, 2003) and these are assessed with multiple goodness-of-fit tests, including Cramer-von Mises test (e.g. Genest et al., 2009), BIC, Akaike Information Criterion (AIC), maximum likelihood, Nash-Sutcliffe efficiency (NSE), etc. (Sadegh et al., 2017). The "AND" hazard scenario is chosen to
consider compound drivers from the joint extreme values of both drivers (Salvadori and De Michele, 2004; Ward et al., 2018; Moftakhari et al., 2019; Couasnon et al., 2020).





For inter-comparing probabilities, we select one copula model that is suitable across all event types (TC, ETC, All and Neither). Instead of pursuing the copula that leads to the best goodness of fit metric for each event type, we use the same copula for all to avoid differences arising due to differing copula models. For example, we avoid intercomparing probability (or return period) results where one event type has been fitted with an extreme value copula and another with a non-extreme value copula. The $p$ values from the Cramer-von Mises test are used to eliminate the inadmissible copulas ($p<0.05$) (Genest et al., 2009; Mazdiyasni et al., 2019; Lucey and Gallien, 2022). BIC is commonly used in previous studies among the metrics for goodness-of-fit, because it considers both sample size and the complexity of the model to avoid overfitting (e.g. Bevacqua et al., 2019; Tootoonchi et al., 2022). Hence, we use BIC as the primary judgment for suitability checks among all the plausible copula models. The Plackett Copula is the most consistently highly ranked copula based on BIC across all event types. The Plackett Copula is flexible in modelling various types of dependence structures and can exhibit tail dependence as well (Nelsen, 2006); it was found here to be suitable across all cases (Table S1).

## 4 Results

Results below include the relative frequencies of top-ranked pluvial and coastal events by storm type (Section 4.1), the measures of dependence of the rain and NTR (Section 4.2), and the magnitude of the marginals of the rain and NTR (Section 4.3). The lag time between the rain and NTR flood drivers are evaluated in Section 4.4. The above first three aspects are dominant factors that influence the joint probability analysis results in Section 4.5.



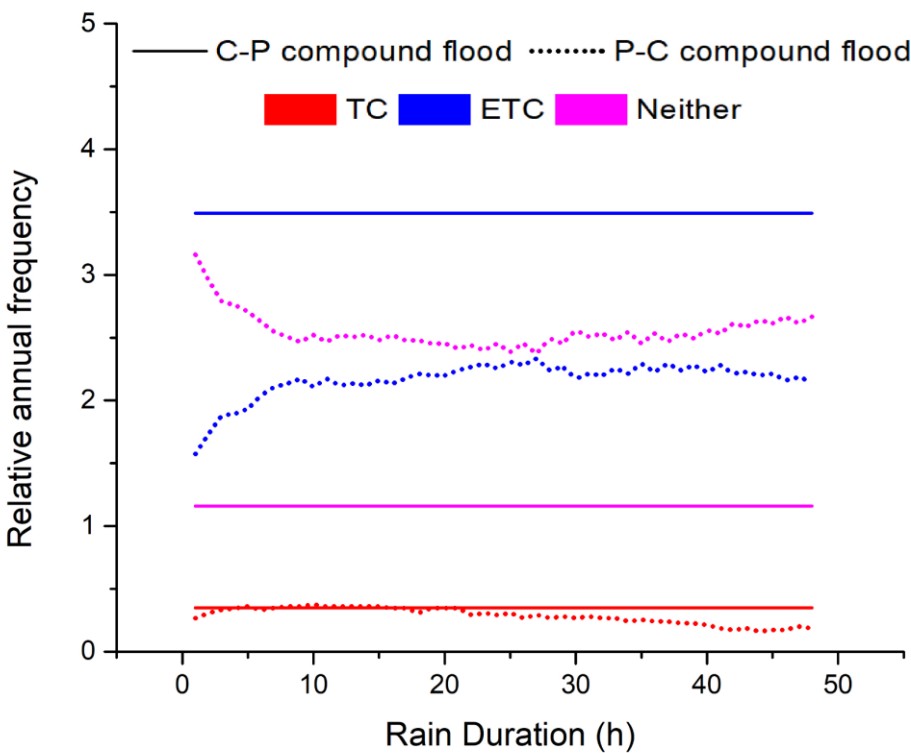

**Figure 3: Compound events at the Battery associated with different storm types and primary drivers. The relative annual frequency**
**conditioned on total rainfall across different durations are represented by dotted lines (P-C compound event) and conditioned on**
**hourly peak NTR are represented by solid lines (C-P compound event). The latter are always flat because the ranked list of NTR (a**
**peak) is the same regardless of rain duration, while the former vary because each duration has a different ranked list.**

## 4.1 Relative Frequency by event types

Figure 3 gives a picture of the average annual frequency for the top-ranked coastal and pluvial events at the Battery associated
with different storm types above a total 5 PYA threshold from all events (i.e. both NTR and rain exceed their respective PYA
thresholds). At Kings Point, the frequencies have similar patterns. Events associated with TCs contribute a small proportion
of all top ranked P-C compound events (0.20-0.40 PYA) and C-P compound event (0.35 PYA) events. Events associated with
ETCs are the major proportion of the top ranked C-P compound events, while neither drive the majority for the top ranked P-
C compound events. For the P-C compound events conditioned on short durations (1-6 hours) rain, a higher proportion is
associated with Neither (e.g. summer convective-rain storms) than for those conditioned on longer durations. These convective
events normally do not last long and typically have a short duration of intense rain.





### 4.2 Dependence

#### 4.2.1 Top-ranked rain intensity of different durations

Using the common method of assessing correlations of storm-duration maximum flood drivers, we find for TCs a large

difference between correlations of P-C and C-P events. TC-induced P-C and C-P compound events at NYC have quite different rain-NTR correlations of 0.3-0.5 and 0.0-0.1, respectively (Fig. S1). On the other hand, the correlations for All, Neither and ETC are ~0.1 for both P-C compound events and C-P compound events. Their correlation is not sensitive to the durations of maximum rain accumulations, except for the short durations (1- 5 hours) which have ~0.0 correlation for P-C compound events. Among the short duration high rain events, a higher proportion of them are in the Neither category that presumably

induced by convective events (Fig. 3). These events have a lower chance of co-occurring with a coastal flood. The P-C correlations are relatively sensitive to the short rain durations (1- 5 hours) associated with both TCs and ETCs (Fig. S1a). Short extreme rain events tend to have high intermittency, leading to high variation in the correlation from one duration to the next. The storm-duration maximum compound events could have maximum flood drivers sequentially or simultaneously, depending on the lag time (explored in Section 4.4) of the compound drivers. So, their compound effects on a pluvial environment like

NYC has a large uncertainty. In the next Section 4.2.2, we investigate the dependence of simultaneous compound flood drivers.

#### 4.2.2 Simultaneous hourly flood drivers

Figure 4 shows the P-C simultaneous compound flood drivers associated with TCs have much higher overall correlation than the other storm types. This pattern is consistent across space in terms of overall dependency (Fig. 4) and upper tail dependence (Fig. S2). This implies a higher joint risk for TCs compared with other storms in terms of dependence.

However, for TCs the C-P simultaneous correlation is near zero at both the Battery and Kings Point. Even if we only look at the upper tail region, the C-P compound events are still less correlated than the P-C compound events driven by TC (comparing panel a and b in Fig. S2). The upper tail dependence coefficients associated with TC are close or slightly higher than other storm types for the majority of locations (e.g. EWR, LGA or the spatial average in Fig. S2).

For certain stations (e.g. JFK, LGA in Fig. 4), the C-P simultaneous compound hourly flood drivers associated with ETC or

All have moderate dependency, but they are not evident in the upper tail region (Fig. S2). At LGA and EWR, their upper tail correlations are less than that associated with TC. The rain-NTR dependency could be significantly different depending on storm type as well as the choice of primary flood driver. Location and rainfall accumulation duration also cause minor changes in correlation. The phenomena are not explored in detail in prior studies in this area. More comparisons are discussed in Sections 5.2.



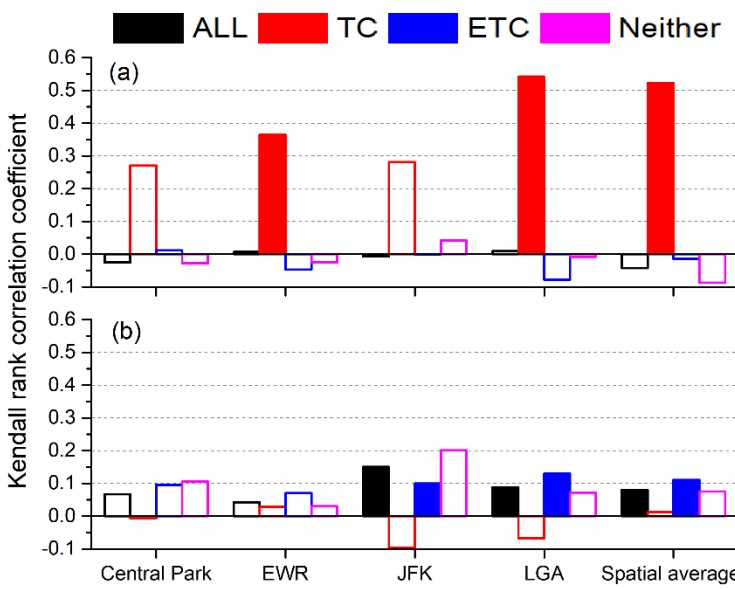


**Figure 4: The Kendall rank correlation coefficients for (a) the P-C simultaneous compound flood drivers and (b) C-P simultaneous compound flood drivers associated with different storm types for each single gauge and the spatial average around the Battery. The filled colour bars represent statistically significant cases ($p<0.05$).**

## 4.3 Magnitudes of flood drivers

The hourly magnitudes of the primary flood drivers are similar for different storm types, but the magnitude of secondary flood drivers associated with TCs stands out compared with other storm types. In Fig. 5, for the P-C compound event, the empirical percentiles (50th and 90th) of top ranked rainfall are slightly higher for TCs than the other storm types, while the empirical percentiles (50th and 90th) of the associated NTR are much higher for TCs than the other types (panels a6 vs a5, a7, a8). Similarly, for the C-P compound event, the empirical 90th percentiles of top ranked NTR are much higher for TCs than the

other types, but the 50th percentiles are similar for all cyclone types. The empirical percentiles (50th and 90th) of the associated compound rainfall are much higher for TCs than the other types (panels b6 vs b7&8). This higher magnitude of the secondary flood driver associated with TCs is not only seen during the simultaneous peak rain or peak NTR hour (at "0" hours in Fig. 5), but also during the few hours around it.

The Battery and Kings Point have similar magnitudes of coupled flood drivers for different storm types. The pattern for the

magnitude of secondary flood drivers described above is also true for Kings Point (Fig. S3). However, the magnitude of the secondary flood drivers for the Battery (panels a5-a8 and b5-b8 in Fig. 5) is relative symmetrically distributed temporally around the peak hourly rain (P-C compound event) or the peak hourly NTR (C-P compound event), while the temporal distribution at Kings Point is asymmetrical (panels a5-a8 and b5-b8 in Fig. S3). Regardless of storm type, the hourly peak





magnitudes of the coupled flood drivers at the Battery are almost simultaneous, while they lag by a few hours at Kings Point.

Even though there is still a simultaneous compound effect at the Kings Point, the magnitude of the simultaneous secondary flood drivers tends to be lower than that of the Battery.



**Figure 5: The magnitude of the P-C compound flood drivers (top; a1-a8) and the C-P compound flood drivers (bottom; b1-b8) by different storm types for the Battery. The top row (a1-a4) and third row (b1-b4) show the primary flood driver magnitudes (50th and 90th percentile), and the second row (a5-a8) and bottom row (b5-b8) show the secondary flood driver magnitudes. X-axis ranges from -10 to 10 h indicating time relative to the peak hour of rain (a1-a8) or NTR (b1-b8).**

## 4.4 Lag time of flood drivers

Figure 6 shows a large proportion of the historical compound events have their maximum coastal NTR after their maximum hourly rainfall, no matter what storm type, primary flood driver or location (Battery or Kings Point). This could arise if rainfall precedes a storm or if the propagation of storm surge into the harbour from offshore is slower than the storm speed (e.g. Orton et al., 2012).

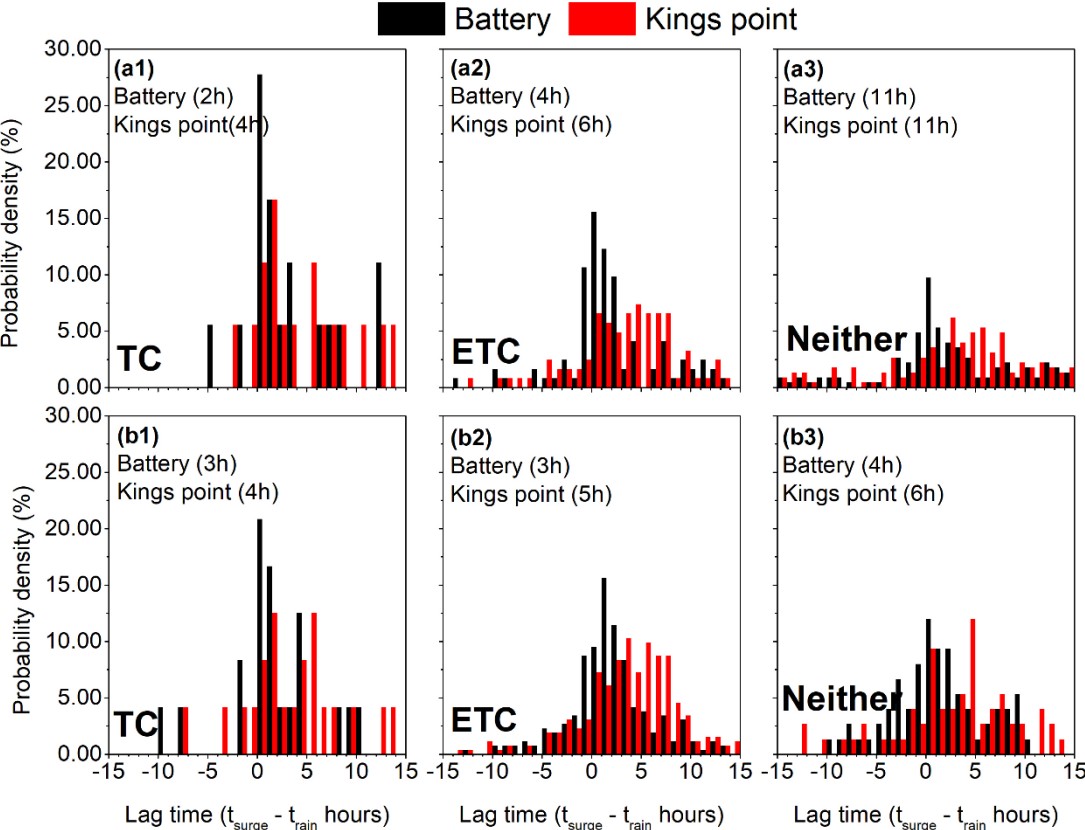

**Figure 6: Histograms of lag time for the P-C compound events (top) and the C-P compound events (bottom). The left, middle and right panels are associated with TC (1), ETC (2), and Neither (3). The red and black colours represent histograms for the Battery and Kings Point separately. The numbers in each plot are the median of the absolute lag time for Battery and Kings Point. Positive lag time values indicate peak storm surge occurs after peak of rainfall. Here we only show the lag time within ±15 hours.**





The statistical characteristics of lag time are different across the three storm types (Fig. 6). During TCs, the secondary maximum flood drivers tend to be either simultaneous or occur with a smaller lag time than for other storm types, which can be seen by comparing histograms a1 and b1 with histograms a2&3 and b2&3 in Fig. 6. The median lag time for TCs tends to

be the smallest and is shown as the parenthetic numbers at top left of each panel. Also, there is a significant negative correlation of the extreme rainfall and the absolute value of lag time during TCs for P-C compound event (Fig. 7). This indicates that the most intense rainfall events tend to have the shortest absolute lag times to the peak, which raises the risk of amplifying the compound flood effects during TCs. These negative correlations are significant at many stations around the Battery (Fig. 7) and Kings Point.

For the Neither type (panels a3 & b3 in Fig. 6), the lag time for P-C compound event (panel a3 in Fig. 6) is more spread out than C-P compound event (panel b3 in Fig. 6). This may be because C-P compound event has storm surge and thus is often associated with a synoptic storm. P-C compound event in the Neither category is more likely not to be associated with an organized storm that produces surge, so there is less reason for timing to be coupled.

The histograms and median absolute lag time also show that the lag time around the Battery tends to be shorter than that around

Kings Point. This is consistent with the magnitude results in Section 4.3. This phenomenon will be further discussed in Section 5.3.

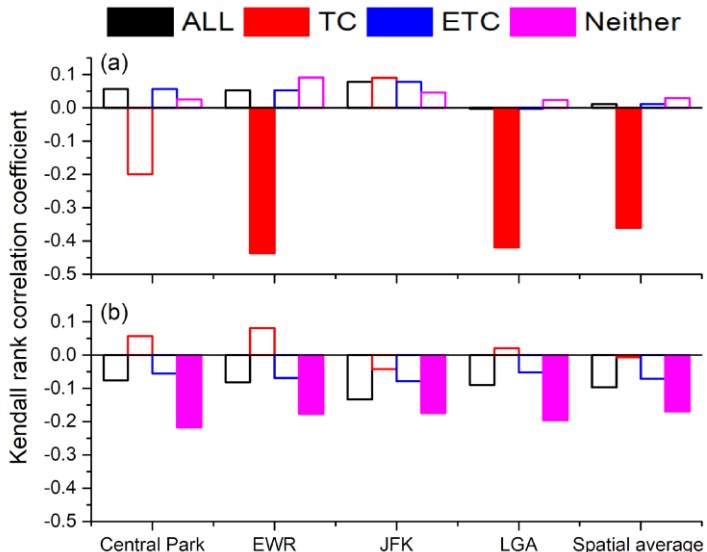

**Figure 7. The Kendall rank correlation coefficients between primary flood drivers and absolute lag time of the secondary driver for (a) the P-C compound event (b) C-P compound event associated with different storm types for each single gauge and the spatial**

**average around the Battery. The filled bars represent statistically significant cases ($p<0.05$).**





## 4.5 Joint return period analysis

In Fig. 8, we contrast the resulting JRP curves associated with each storm type with that evaluated from All. For P-C events, the analysis uses simultaneous NTR. For C-P events, the analysis uses rainfall within a ±1 hour window from the time of peak NTR. This is done to conservatively assess the joint occurrence, because rainfall during severe storms can sharply change from one hour to the next. Extreme surge typically lasts for several hours at minimum (Booth et al., 2016), whereas extreme rain can last for only an hour or less and then drop to zero or can abruptly alternate between extreme and zero when there is banding of rainfall. This is demonstrated in Fig. 8, comparing the sharpness of peaks in panels a6 and b6. For P-C events (a6) the simultaneous hourly NTR barely drops for lags of ±1 hour. For C-P events (b6), the 90th percentile simultaneous rain on average drops to ~40% in the hour after peak NTR. For the 50th percentile, the average drop is ~50%. The need for the ±1 hour window is also illustrated by the fact that the peak for the 50th percentile rain (panel b6) is at -1 hour.









**Figure 8: Observations (black points) and joint return period curves (5-year to 200-year) for the P-C compound event (left; a1-a4) and the C-P compound event (right; b1-b4) by different storm types. All the compound flood drivers are hourly and simultaneous, except maximum rainfall within a ±1-hour window was used for C-P events (see section 4.5).**


The JRP results show that TCs play a dominant role in driving the most extreme (50-year return period and above) compound events, while ETCs contribute mainly to the more frequent compound events (10-year events and below). Specifically in Fig. 8, for the joint extreme values of the 50-year (or above) return periods, higher values occur for TC than for All, but for the joint extreme values of the 10-year (or less) return periods, the joint values of TCs are much smaller than that of All.

Conversely, the ETC joint extreme values of the 10-year (or less) return period are similar to All, but much smaller than TC for the 50-year (or above). The values for Neither are lower than for the other storm types, and thus play a very limited role. The characteristics of the JRP curves relate to both the rank correlations and marginal intensities presented in Sections 4.2 and 4.3. The P-C joint return period (JRP) curves associated with TCs, for example, are more convex than those associated with other storm types, as they have a higher dependency. For the C-P compound event, the JRP curves associated with TCs are

less convex than the P-C compound event, due to low correlations. However, they still cause stronger extreme compound events than other storm types. While TCs are far less frequent than other storm types (Section 4.1), they are the primary source of extreme compound rain-surge (e.g. 50-year to 200-year events).

## 5 Discussion

Our results illustrate that events driven by TCs, ETCs and Neither can have significant differences in their compound flood

hazard characteristics. Separating the data from different storm type results in different estimates of their dependency, magnitudes of marginals, lag time and occurrence frequency. TCs have markedly different driver characteristics from other storm types and dominate the joint probabilities of the most extreme rain-surge compound events, even though they occur much less frequently. ETCs are the predominant source of more frequent, moderate compound events. Critically, the 50-year to 200-year return periods for compound events are higher when only assessing TC versus assessing All events together. This

is mainly due to the larger magnitude of the secondary flood driver (Section 4.3) and in the case of P-C events also a higher dependency (Section 4.2). This research demonstrates that the danger of compound hazards from TCs can be underestimated if aggregated with data with other storm types, which is common practice. This is discussed in more detail below in Section 5.1.

Prior research on pluvial-coastal compounding typically uses the daily rainfall data and ±1 day window to capture the

compound flood drivers (e.g. Lai et al., 2021; Kim et al., 2023) and evaluate the joint return periods using All events (e.g. Ghanbari et al., 2023; Wahl et al., 2015). However, small lags of 0-2 hours between maximum rainfall and coastal drivers were found to be a critical factor on the magnitude of urban compound flood impacts (e.g. Gori et al., 2020a). The analysis of hourly data verifies that rain and NTR occur simultaneously and also opens up many new windows into potential compounding that





would be missed if using daily rainfall data and the storm-maximum approach. Results show the compound hazard statistics
using the hourly simultaneous approach can reveal smaller hazard extremes from those resulting from the more commonly
applied "storm-duration maximum" approach (Fig. S4), especially for the location (Kings Point) with more prominent lag
times (Fig. 5 vs Fig. S3, or Fig. 6). Storm-duration co-occurrences of extreme flood drivers can be a day or more apart, so can
have sequential or compounding impacts, whereas the hourly simultaneous approach guarantees that the drivers coincide. The
"hourly simultaneous" analysis results can also be different for nearby tide stations due to the direction and pathway of storm
surge propagation. This phenomenon is discussed in Section 5.3. Limitations and simplifications of the research are discussed
in Section 5.4.

## 5.1 Storm Separation

TCs, ETCs and Neither can be responsible for similar individual events, but their clouds of data points and copula-modelled
JRP curves for rain and NTR are often distinct (Fig. 8). TCs and ETCs both can trigger extreme coastal flooding and extreme
rainfall, whereas the Neither (often summer convective thunderstorms) mostly only cause extreme rainfall.

The decision to separate rare TCs from other more frequent storm event types in extreme value analyses is a difficult one, as
it results in more uncertainty in probability distributions. This can be a challenge for policy-oriented metrics such as the 100-
year flood zone, especially if it leads to widely varying estimates across different assessment methods or for consecutive
studies from one organization (e.g. FEMA; Orton et al., 2016). Nevertheless, for environments like NYC where TCs are
infrequent but responsible for a majority of historical rain and storm surge extreme events, merging data in an analysis of All
storm event data can often lead to low biases in probabilistic assessments. If TCs are separated, the results in Figure 8 show
that additional separation of ETCs from Neither may also be beneficial for proper quantification of 10-year return period joint
rain-surge events.

Recent studies (e.g. Gori et al., 2022) have begun using synthetic TC storms to evaluate the joint probability of compound rain
and coastal flood hazards, which can be particularly useful for assessing future climate change. The observation-based
approach we have used, and this new model-based approach can be complementary. While the observation-based approach is
grounded in real-world data, the model-based approach can improve sample sizes for extreme events and enable extending the
science to climate projections.

## 5.2 Different correlations for C-P and P-C associated with TCs

NYC C-P compound hazard induced by TCs has a much lower correlation than P-C compound hazard, in contrast to prior
research for Florida and Texas. Typical TCs have their heaviest rain and strongest onshore winds (and thus, surge) in separate
quadrants (Yang et al., 2021), and as a result rain and surge are not typically highly correlated. Typical correlations for Florida
and Texas are 0.2-0.4 for both P-C and C-P events (Jane et al., 2020; Kim et al., 2023). For top-ranked NYC NTR (C-P) events,
the correlations are near zero. One reason this could be the case is because the TCs at this high latitude are typically undergoing
extratropical transition causing the rain to get even more separated, moving radially outward further from the storm centre





(e.g. Evans et al., 2017). For example, during Hurricane Sandy, the storm size became larger as the storm transitioned and the distance between rain and surge became large. Storm surge was concentrated near the right side of the storm track in the region of highest onshore winds (New York Bight), and precipitation was radially outward and to the west of the storm (e.g. Virginia, West Virginia; Blake et al., 2013). Regardless of the lower or even negative correlations between NTR and rain, there are still

stronger secondary driver magnitudes and higher joint return period curves (50- to 200-year return periods) during TCs than All events.

### 5.3 Storm track and surge path dependence of compounding

The results by storm type reveal characteristics of compound flood drivers could be dramatically different depending on the storm type association. Similarly, for one specific storm type, compound hazards risk may mainly come from events with

certain cyclone tracks. For example, NYC has exposure to three main hurricane paths pertinent to surge: (1) New Jersey landfalls, which maximize storm surge "to the right of the storm" at NYC but often co-occurs with low rainfall, as the rain tends to occur toward the west (e.g. Sandy), (2) direct hits from the south bringing large surges and heavy rains, and (3) tracks crossing Long Island to the east, where surges travel westward across Long Island Sound and there is also potential for heavy coincident rainfall to the "left" of the track. An initial hypothesis of this research was that Kings Point, due to events with track

type (3), would have higher compound rain-NTR hazard than the Battery. However, we find mixed evidence that is not strongly supportive of this hypothesis. First, rank correlations for both stations were very similar. For P-C compounding, Kings Point has higher 90th and 50th percentiles of NTR than Battery (Fig. 5, Fig. S3), supporting the hypothesis. However, for C-P compounding, Battery has higher 90th percentile rainfall than Kings Point.

A complexity not explored here is that different storm types have different track paths. ETCs have tracks from both seaward

(the south) and landward (Booth et al., 2016), while TCs do not come over land or more precisely weaken and convert to post-tropical status (e.g. Ida) or dissipate when they do. Compound flood characteristics of these different storm types could stratify depending on the tracks. More research on the storm track dependency of compound flood hazards would be useful.

The surge path is also an important factor that could affect the timing of compound flood drivers, which could change the compound effects and risk locally during storms. For example, for an urban pluvial environment like NYC. The Battery and

Kings Point have qualitatively similar storm-duration maximum rain-NTR compound hazard characteristics. However, we found the peak NTR at Kings Point tends to have longer hours of lag time from its peak rain during TCs and other storm events, due to surge propagation along Long Island Sound. This could reduce the risk of pluvial-coastal compound flood hazards but raises the risk of fluvial-coastal compound (e.g. storms in Table 1 in Chen et al., 2020). Examining river stage data (USGS station #01302020) for post-TC Ida, we see a 2-hour lag time between the onset of heavy rainfall and the exceedance

of the Major flood stage, with river stage remaining high for 18+ hours afterward. Given the typical lag time to peak surge for TCs of 2-6 hours for Kings Point (red histogram of panels a1 and b1 in Fig. 6), this could lead to an elevated risk of compound fluvial-coastal flooding.



### 5.4 Limitations and simplifications

Some limitations or challenges of our study are noted herein. First, a few C-P compound events associated with Neither have zero rainfall, which causes anomalies (Kojadinovic and Yan, 2010) when modelling its marginal distribution and dependency with NTR. We found that there are only slight changes in the correlation coefficient when omitting these few events. Nevertheless, neglecting them could cause a small negative effect on the joint probability results (Panel b4 in Fig. 8). The joint probability curves for Neither are not the key result for this research, so we did not apply a more sophisticated approach (e.g.

randomization techniques in De Michele et al., 2013) to improve this issue. Secondly, tide is a relatively uncorrelated component in the total water level. Our main research interest is to investigate the statistical characteristics of the joint rainfall and storm surge. We choose to use the NTR, instead of water level, as the values for coastal hazards to avoid the interference of the randomness of tide (Bevacqua et al., 2019; Jane et al., 2022; Paprotny et al., 2018; Wahl et al., 2015). Future analysis could include tide as another driver of coastal hazards assuming tide as an independent component that could be near linearly

superimposed with the NTR at this location (e.g. Jordi et al., 2018). Especially for those areas (e.g. coast of Jamaica Bay) already suffering from nuisance flooding (Orton et al., 2015) due to low elevation, there could be potential compound nuisance floods dominated by rainfall and high tide (e.g. Sept. 29, 2023 flood around Flushing bay with moderate rainfall, high tide and no storm surge).

Lastly, this study does not evaluate historical or future climate change but focuses instead on establishing a baseline assessment

of rain-surge compound hazard. We use past data to look at the present compound flood risk, requiring an assumption that the past processes and probabilities reflect those of the present. We remove sea level rise from the storm surge data in order to eliminate the most well-established climate change effect. While the broader Northeastern U.S. region has seen an increase in rainfall coming in extreme events (Huang et al., 2021), no observational study has revealed increases in extreme rainfall for New York City. One study has shown increasing storm surges and storm tides from 1844-2013, but those increases occurred

in the century leading up to the 1950s (Talke et al., 2014). Extremes of both rainfall and storm surges were not found to have significant increases during the historic period that we evaluate in our research (1948-present; Wahl et al., 2015). So, it is reasonable for this first baseline assessment that we assume that rainfall and storm surge are statistically stationary.

### 6 Conclusions

Flood risk studies, insurance products and flood maps typically assume rain and storm surge are independent processes.

However, for NYC our research shows non-zero correlations between these flood drivers and that there is a higher probability of one variable being extreme when the other is extreme. Based on 75 years of historical observations, compound rain and NTR overall have a low, but non-zero rank correlation (~0.10-0.15). However, the dependency of compound rain and NTR associated with TCs alone can be high. In addition, the magnitudes of secondary flood drivers during TCs are much higher compared with other event types. The lag time between the compound flood drivers also differs by storm type, with TCs

tending to have the lowest absolute lag time compared with ETCs or Neither (convective storms and other types of events).





TCs also tend to have more simultaneous occurrence with NTR as the rain intensity rises. In toto, this evidence suggests that TC events need separate assessment, to avoid underestimating compound flood risk.

The Battery and Kings Point coastal areas of NYC have qualitatively similar compound rain-surge hazard correlations. However, we found the peak NTR at Kings Point tends to lag hours behind the peak rain for all storm types, likely due to

propagation of the storm surge along Long Island Sound. The timing between the compound flood drivers is a critical factor that affects their compound effects, especially in terms of an urban pluvial environment like NYC. This lag could reduce the risk of pluvial-coastal compound flood hazards but may raise the risk of fluvial-coastal compound floods.

The historical data analysis shows that the combination of extreme rain and extreme surge (e.g. Hurricane Irene 2011 -50yr; Hurricane Gloria, 1985 -200yr) generally has a low annual probability in NYC. However, these statistical results are only

based on the limited number of TC events (roughly 0.3 events per year) that hit NYC. NYC's extreme events often cause only one extreme flood driver. For example, Hurricane Sandy (2012) triggered an extreme storm surge in NYC with only moderate rain, and Hurricane Ida (2021) triggered extreme rainfall but went through the ETC transition with less wind and only a small storm surge to the right of the storm track. However, our joint probabilistic analysis demonstrates that TCs have the potential to trigger both extremes at the same time, potentially causing a major flood disaster due to the non-linear increase in impacts

with flood magnitude. While TCs are far less frequent than other storm types, they are the primary source of compound rain-surge extremes (e.g. 50-year and 100-year events).

Statistical and probabilistic assessments of rain and storm surge such as this demonstrate that flood drivers can co-occur during extreme storm events. Furthermore, statistical analysis choices between hourly and daily data, and rules for storm duration maxima, may be debated. Co-occurrence does not guarantee additive behaviour where flooding is actually compounded.

Therefore, an important next step will be to simulate these extreme event scenarios in flood models such as those described above. Given the availability of one or more such flood models (e.g. Ghanbari et al., 2023), it is recommended that an assessment of the on-the-ground impacts of these compound events is initiated.

**Code availability**

The codes for the data are primarily available at Chen and Orton (2024). One exception is the MATLAB copula toolbox which

is available from the URL https://amir.eng.uci.edu/MhAST.php (Sadegh et al., 2018).

**Data availability**

The hourly rainfall data is from NOAA National Climatic Data Centre (NCDC) https://www.ncdc.noaa.gov/cdo-web/datasets and the Automated Surface Observing System (ASOS) https://mesonet.agron.iastate.edu/ASOS/. The water level data is available at https://tidesandcurrents.noaa.gov/stations.html?type=Water+Levels. The TC track data is obtained from the

National Hurricane Centre (NHC) https://www.nhc.noaa.gov/data/.





**Author contributions**

ZC PO JB and TW conceived the research. PO acquired the funding and administrated the project. ZC performed the data analysis. TW contributed to the Copula method. ZC and PO prepared the manuscript draft. All authors revise the manuscript.

**Competing interests**

The contact author has declared that neither they nor their co-authors have any competing interests.

**Disclaimer**

Publisher's note: Copernicus Publications remains neutral with regard to jurisdictional claims made in the text, published maps, institutional affiliations, or any other geographical representation in this paper.

**Acknowledgments**

The authors would like to thank constructive instructions from Moji Sadegh (Boise State University) on the MhAST. We also thank James Booth and Max Sehaumpai (City University of New York) for sharing their reanalysis ETC track data. We acknowledges the copyright ownership of Esri (the company behind ArcGIS) for the software used to create the map in Fig. 1.

**Financial support**

This work is supported through a consortium contract, "Climate Vulnerability, Impact, and Adaptation Analysis" with the NYC Department of Citywide Administrative Services and the Mayor's Office of Climate and Environmental Justice and the National Oceanic and Atmospheric Administration Regional Integrated Sciences and Assessments/Climate Adaptation Partnerships (RISA/CAP) award NA21OAR4310313. T.W. acknowledges support by the National Science Foundation (grant numbers 1929382 and 2103754) and the USACE Climate Preparedness and Resilience Community of Practice and Programs.

R.M.H acknowledges support from the NSF/COPE MACH project. A.T.D. acknowledges support from NOAA contract NE-EN6100-23-00822.

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
