# Peer review of "Influence of Storm Type on Compound Flood Hazard of a Mid-Latitude Coastal-Urban Environment"

_Hydrology and Earth System Sciences, 2024_

## Author Comment (AC1)

**Reviewer 1:**

The article "Influence of Storm Type on Compound Flood Hazard of a Mid-Latitude Coastal-Urban Environment" explores the differences between tropical cyclone (TC), extra-tropical cyclone (ETC) and non-cyclone (NC) as drivers of compound flooding to New York city, USA. The authors examine historic time-series of hourly rain and tide gauge data, using dependence and joint probability analysis methods, to explore the potential influence of storm type on near-simultaneous pluvial and storm surge flooding events. The study found that TCs dominate the most extreme pluvial/storm surge compound flood events, but ETCs are responsible for the majority of moderate and high frequency occurrences. There are important magnitude and lag differences depending on coastal location.

**Strengths:**

1. **Aim and objectives:** the aims and objectives of the study are clearly stated. The use of long time-series of gauged data is to be welcomed.

2. **Discussion and conclusions:** the focus on compound floods due to TC- and ETC-linked drivers is timely, given the rising frequency of extreme weather even within temperate zones, due to climate change.

3. **Statement of limitations:** it benefits this study that simplifications and assumptions are clearly stated. This text provides context and is a source of ideas for future research.

The authors would like to thank the Reviewer for providing thoughtful comments. Below, the Reviewer can find our responses to each comment, including how we will address each of them in the revised manuscript.

*Minor issues:*

1. *Sea level rise trend: it was not completely clear how the trend of sea level rise has been removed from the 75-year time-series data measured at the tide gauges. Could the authors please expand on this?*

   We appreciate the reviewer's comment. We will clarify this methodology in Section 3.2 and include additional details to ensure reproducibility.

2. ***Distinction between TCs, ETCs and NCs/convective storms:*** *It was not clear to me how the authors categorized the different storm types in section 2. Was this pre-assigned to each storm by the National Hurricane Centre, or was a threshold (e.g. as defined by the Saffir-Simpson scale) applied afterwards? This is key information for anyone wishing to reproduce the study.*

   The categorization of storm types (Tropical Cyclones (TCs), Extratropical Cyclones (ETCs), and "Neither") was pre-assigned using records from the National Hurricane Center (NHC) and the reanalysis data (Section 2.2.3). Storms in the HURDAT dataset were considered TCs, given that they are nearly always either TC or post-tropical cyclones. It is very unusual for an extratropical cyclone to transition to tropical before passing NYB, with one case being the 1991 perfect storm.

   We will expand Section 3.4 to explain this classification process in greater detail.

3. ***POT approach:*** *could the authors please expand on the selection of the top-5 ranked rain/surge events each year? For context, it would be interesting to know how many of these events (out of all 75 x 5 events picked, per gauge), were categorized TCs, ETCs or NC/convective storms. A simple table would be enough. Only being able to capture a few TC events in the record, even with a long time-series, has to be recognized as an unavoidable limitation.*

   We agree with the reviewer. First, we already have Figure 3 to describe the annual frequency of each storm type within the top-ranked events for the Battery station. The selection of the top-ranked rain/surge events per year was based on their magnitude using the Peaks-Over-Threshold (POT) approach, representing an average annual exceedance frequency of 5/year (not the same as "top-5" per year).

   Out of the total 75 × 5 events across the 75-year dataset per gauge, we will add a new table summarizing the breakdown of TCs, ETCs, and Neither for both the Battery and the Kings point stations in the supplementary material.

   This table will highlight the relatively low frequency of TCs due to their rarity. In addition, we will add more text in the limitation section of the paper to acknowledge the challenge of capturing these events even in a long time series.

4. **Return period assumptions:** *the data are not longer than 75-years, how do the authors defend the calculation of return periods in excess of this (e.g. in figure 8)?*

Thank you for noting this point.

Due to the importance of extreme events (e.g. 100-year return levels) in planning and insurance, using shorter durations (e.g. our 75 years -1948 to 2022) of observed data to predict return periods in longer periods is common for univariate (e.g., Arns et al., 2015) and bivariate (e.g., Zscheischler et al. 2017) analysis.

In Table S1, the $p$ values demonstrate that the copula models and thus return periods are plausible. Also, the scatterplots in Figure 8 also qualitatively back up our conclusion that TCs lead to worse joint hazards than other storm types. Importantly, we are not citing the exact values of rain and surge pertaining to specific return periods, we are only showing that an analysis of "ALL" data leads to smaller joint 50- to 200-year return levels for both flood drivers than analysis of TCs only. Therefore, our conclusions are robust to uncertainties in the fitted copula models.

5. **Assumption of stationary storm surge over time:** *While it is stated that this first baseline assessment does simplify conditions, it could also be worth mentioning that recent research has identified that elsewhere, storm surge extremes are not in fact stationary, over similar time-scales (e.g. Calafat et al 2022, DOI: https://doi.org/10.1038/s41586-022-04426-5 )*

We agree this is worth mentioning and will add text mentioning the Calafat paper in the Discussion section on storm stationarity assumption.

6. **Figures:** *the majority of figures would be difficult to read for those who print in B&W, or are color-blind. Would recommend a different color palette (colorbrewer2.org, for example, suggests great color combinations that overcome this problem). In addition, would suggest that Figure 1 would benefit from (a) a simpler (line drawing) background rather than satellite imagery; (b) an inset, or wider view, to illustrate the NYC location with more of the Long Island Sound and Atlantic visible (to better understand storm surge, and significance of storm orientation at each gauge location); and (c) to perhaps reconsider the color scheme of gauge location points for the reasons stated to above. Additionally, figure 8 would benefit from larger font in the x-, and y- axis labels.*

We appreciate the feedback on figure accessibility and will address these concerns:

(a) We will simplify the background of Figure 1 to a line drawing for clarity.

(b) We will add an inset or wider view in Figure 1 to provide context for the NYC location and surrounding regions.

(c) We will revise the color scheme for all figures to use colorblind-friendly palettes (e.g., from ColorBrewer2.org).

(d) We will increase the font size of axis labels in Figure 8 for readability.

7. **Statement of relevance:** *the manuscript might benefit from a clearer description of the significance of the results of this study, which focuses on a relatively small urban watershed referencing a small number of gauges, to the current scientific knowledge of pluvial/coastal compound flooding. How do these findings contribute to the scientific conversation?*

We agree that the primary contributions were not coming through clearly. To improve this, we will enhance the start of the Discussion section to better articulate the broader implications of our findings. The first paragraph already presents the value of our separation of TCs and other storm types, which has rarely been done for compound hazard research. The second paragraph already presents the value of using hourly data and the new knowledge that is obtained which would not be possible using past common methods of daily data and loose definitions of overlap. For both paragraphs, we will add topic sentences that make clear how these two core approaches of our paper contribute to the broader field.

**Overall:**

Because of the use of hourly time-series data, this study provides useful insights into how lag time, magnitude, and orientation of storm-linked drivers all contribute to the state of flooding within an urban watershed of high economic value. The use of this more discrete data, creates a useful distinction between impacts in compound flooding due to TCs, ETCs, and convective storms. The study would benefit from clarifying some details of the methodology and results, as detailed above.

**Technical corrections:**

- *L486 "in toto"?*

  Thank you! It is a typo. We will replace it with "in total" to improve readability.

- *L114 - how long is the data collected at Battery gauge?*

  The text will be changed to clarify that the Battery gauge has near-complete long-term temporal coverage during the period of hourly rain data from 1948 to 2022, spanning approximately 75 years.

- *L 143 – what is a "sewershed"?*

  A "sewershed" refers to an area of land where all surface water drains into a common sewer system, similar to a watershed but specifically for urban stormwater and wastewater systems.

- *L165-L167. At a single gauge is this statement correct? This feature of storm surge is known due to onshore and offshore winds in different quadrants of the TC position; however usually one tide gauge records rising levels due to onshore winds, and a neighbor some km away would (hopefully be well-placed to) capture the negative surge due to offshore winds. Of course this effect changes with cyclone path/coastline orientation, and cyclone size.*

  We agree that the text in this paragraph was confusing, and felt that it was not needed.  We will eliminate the statement.  However, we note that there are no known historical cases at NYH where there are such large differences in surge over small distances of a few km.  Surge is typically very similar across NY Harbor, though it can be different at Long Island Sound (Kings Point), which was the reason we separated the analysis into these two areas.

---

## Author Response (AR1)

**Reviewer 1:**

The article "Influence of Storm Type on Compound Flood Hazard of a Mid-Latitude Coastal-Urban Environment" explores the differences between tropical cyclone (TC), extra-tropical cyclone (ETC) and non-cyclone (NC) as drivers of compound flooding to New York city, USA. The authors examine historic time-series of hourly rain and tide gauge data, using dependence and joint probability analysis methods, to explore the potential influence of storm type on near-simultaneous pluvial and storm surge flooding events. The study found that TCs dominate the most extreme pluvial/storm surge compound flood events, but ETCs are responsible for the majority of moderate and high frequency occurrences. There are important magnitude and lag differences depending on coastal location.

**Strengths:**

- 1. Aim and objectives: the aims and objectives of the study are clearly stated. The use of long time-series of gauged data is to be welcomed.
- 2. **Discussion and conclusions:** the focus on compound floods due to TC- and ETClinked drivers is timely, given the rising frequency of extreme weather even within temperate zones, due to climate change.
- 3. **Statement of limitations:** it benefits this study that simplifications and assumptions are clearly stated. This text provides context and is a source of ideas for future research.

The authors would like to thank the Reviewer for providing thoughtful comments. Below, the Reviewer can find our responses to each comment, including how we address each of them in the revised manuscript. The line numbers are based on the **tracked-change version** of the manuscript.

**Minor issues:**

 Sea level rise trend: it was not completely clear how the trend of sea level rise has been removed from the 75-year time-series data measured at the tide gauges. Could the authors please expand on this?

We appreciate the reviewer's comment. We now clarify this methodology in Section 3.2 and include additional details in lines 159-160 to ensure reproducibility.

2. Distinction between TCs, ETCs and NCs/convective storms: It was not clear to me how the authors categorized the different storm types in section 2. Was this preassigned to each storm by the National Hurricane Centre, or was a threshold (e.g. as defined by the Saffir-Simpson scale) applied afterwards? This is key information for anyone wishing to reproduce the study.

The categorization of storm types (Tropical Cyclones (TCs), Extratropical Cyclones (ETCs), and "Neither") was pre-assigned using records from the National Hurricane Center (NHC) and the reanalysis data (Section 2.2.3). Storms in the HURDAT dataset were considered TCs, given that they are nearly always either TC or post-tropical cyclones. It is very unusual for an extratropical cyclone to transition to tropical before passing NYB, with one case being the 1991 perfect storm.

We expand Section 3.4 to explain this classification process in greater detail, ensuring transparency for replication in lines 198-199.

3. **POT approach:** could the authors please expand on the selection of the top-5 ranked rain/surge events each year? For context, it would be interesting to know how many of these events (out of all 75 x 5 events picked, per gauge), were categorized TCs, ETCs or NC/convective storms. A simple table would be enough. Only being able to capture a few TC events in the record, even with a long timeseries, has to be recognized as an unavoidable limitation.

We agree with the reviewer. First, we already have Figure 3 to describe the annual frequency of each storm type within the top-ranked events for the Battery station. The selection of the top-ranked rain/surge events per year was based on their magnitude using the Peaks-Over-Threshold (POT) approach, representing an average annual exceedance frequency of 5/year (not the same as "top-5" per year). Modifications are made in lines 259-261.

Out of the total 75 × 5 events across the 75-year dataset per gauge, we add a new table summarizing the breakdown of TCs, ETCs, and Neither for both the Battery and the Kings point stations in the supplementary material.

This table highlights the relatively low frequency of TCs due to their rarity. In addition, we add more text in the limitation section of the paper to acknowledge the challenge of capturing these events even in a long time series.

4. **Return period assumptions:** the data are not longer than 75-years, how do the authors defend the calculation of return periods in excess of this (e.g. in figure 8)?

Thank you for noting this point.

Due to the importance of extreme events (e.g. 100-year return levels) in planning and insurance, using shorter durations (e.g. our 75 years -1948 to 2022) of observed data to predict larger return periods is common for univariate (e.g., Arns et al., 2015) and bivariate (e.g., Zscheischler et al. 2017) analysis.

In Table S1, the *p* values demonstrate that the copula models and thus return periods are plausible. Also, as stated in the text the scatterplots in Figure 8 also qualitatively back up our conclusion that TCs lead to worse joint hazards than other storm types. Importantly, we are not citing the exact values of rain and surge pertaining to specific return periods, we are only showing that an analysis of "ALL" data leads to smaller joint 50- to 200-year return levels for both flood drivers than analysis of TCs only. Therefore, our conclusions are robust to uncertainties in the fitted copula models.

5. **Assumption of stationary storm surge over time:** While it is stated that this first baseline assessment does simplify conditions, it could also be worth mentioning that recent research has identified that elsewhere, storm surge extremes are not in fact stationary, over similar time-scales (e.g. Calafat et al 2022, DOI: https://doi.org/10.1038/s41586-022-04426-5)

We agree this is worth mentioning and add text mentioning the Calafat paper in the Discussion text on the storm stationarity assumption in lines 502-503.

6. **Figures:** the majority of figures would be difficult to read for those who print in B&W, or are color-blind. Would recommend a different color palette (colorbrewer2.org, for example, suggests great color combinations that overcome this problem). In addition, would suggest that Figure 1 would benefit from (a) a simpler (line drawing) background rather than satellite imagery; (b) an inset, or wider view, to illustrate the NYC location with more of the Long Island Sound and Atlantic visible (to better understand storm surge, and significance of storm orientation at each gauge location); and (c) to perhaps reconsider the color scheme of gauge location points for the reasons stated to above. Additionally, figure 8 would benefit from larger font in the x-, and y- axis labels.

We appreciate the feedback on figure accessibility and address these concerns:

For most figures with several colors or shading, we have revised the color scheme to use colorblind-friendly palettes (e.g., from ColorBrewer2.org).

For Figure 1 (map), We use a wider view in Figure 1 to provide context for the NYC location and surrounding regions. We keep using the satellite imagery as the background because we feel it is helpful to show the urbanization. However, we have kept the color scheme because the stations are marked with different symbols for differentiation.

For Figure 8, we increase the font size of axis labels for readability.

7. **Statement of relevance:** the manuscript might benefit from a clearer description of the significance of the results of this study, which focuses on a relatively small urban watershed referencing a small number of gauges, to the current scientific knowledge of pluvial/coastal compound flooding. How do these findings contribute to the scientific conversation?

We agree that the primary contributions were not coming through clearly. To improve this, we enhance the start of the Discussion section in lines 390-392 to better articulate the broader implications of our findings. The first paragraph already presents the value of our separation of TCs and other storm types, which has rarely been done for compound hazard research. The second paragraph already presents the value of using hourly data and the new knowledge that is obtained which would not be possible using past common methods of daily data and loose definitions of overlap. For both paragraphs, we add topic sentences that make clear how these two core approaches of our paper contribute to the broader field.

**Overall:**

Because of the use of hourly time-series data, this study provides useful insights into how lag time, magnitude, and orientation of storm-linked drivers all contribute to the state of flooding within an urban watershed of high economic value. The use of this more discrete data, creates a useful distinction between impacts in compound flooding due to TCs,

ETCs, and convective storms. The study would benefit from clarifying some details of the methodology and results, as detailed above.

**Technical corrections:**

• L486 "in toto"?

Thank you! It is a typo. We replace it with "in total" in line 513 to improve readability.

**• L114 - how long is the data collected at Battery gauge?**

The text is changed in lines 116-117 to clarify that the Battery gauge has nearcomplete long-term temporal coverage during the period of hourly rain data from 1948 to 2022, spanning approximately 75 years.

• L 143 – what is a "sewershed"?

A "sewershed" refers to an area of land where all surface water drains into a common sewer system, similar to a watershed but specifically for urban stormwater and wastewater systems. We eliminate use of the term, as it was not used elsewhere in the paper.

• L165-L167. At a single gauge is this statement correct? This feature of storm surge is known due to onshore and offshore winds in different quadrants of the TC position; however usually one tide gauge records rising levels due to onshore winds, and a neighbor some km away would (hopefully be well-placed to) capture the negative surge due to offshore winds. Of course this effect changes with cyclone path/coastline orientation, and cyclone size.

We agree that the text in this paragraph was confusing, and felt that it was not needed. We eliminate the statement. However, we note that there are no known historical cases at NYH where there are such large differences in surge over small distances of a few km. Surge is typically very similar across NY Harbor, though it can be different at Long Island Sound (Kings Point), which was the reason we separated the analysis into these two areas.

**Reviewer 2:**

This manuscript by Chen et al., presents an analysis on compound flood hazard for the New York City area. The analysis is focused on compound events of precipitation and storm-surge that are driven by different storm events classified as tropical cyclones (TC), extra-tropical cyclones(ETC) and neither events. Results are also presented for "all" events considered, to highlight the differences in return period of the hazard when frequency analysis does not consider event type. Results shown suggest that despite the fact that the frequency of compound surge and rain events is low, the compound risk associated to TC events need to be assessed separately to avoid underestimation of the risk. Analysis has been based on a long record of hourly rain and tide gauges. Carrying the analysis at an hourly scale offers clear advantages, with respect to past works focusing on daily, on the identification of "simultaneous" rain-surge events and investigation of lag of the peaks (from rain and surge) overall.

Overall, the manuscript is clearly written, and the discussion and conclusions are supported by the results presented. Furthermore, the analysis at hourly scale and the event-type investigation offers novel elements for this type of work. Most of my major concerns on the methodological framework have been acknowledged by the authors themselves in section 5.4 "Limitations and simplifications", a fact that I appreciate because at the very least demonstrates that the authors understand and openly acknowledge the limitations of their approach and the complexity of the problem under study.

The authors would like to thank the Reviewer for their insightful comments. Below are our responses to each comment, including the adjustments to the manuscript. The line numbers are based on the **tracked-change version** of the manuscript.

Below I list some additional comments (mostly minor) for the author's consideration.

1. Thinking of estimation of lag or equivalently identification of "simultaneous" extremes of rainfall and surge, and considering that timing for the two variables is derived from different locations in space (tide gauges for surge and rain gauges for rainfall), there is some potential effect therefore on lag estimation. The authors somewhat refer to this effect in lines 394-395 and mention that this is further discussed in Section 5.3, but it is not

discussed any further in that section. I think that elaborating further on this (and potential implications on the findings or methodology overall) is required.

We thank the reviewer for highlighting this important consideration. We acknowledge that using tide gauge data for surge and rain gauge data for rainfall can introduce spatial mismatches, potentially affecting the precise estimation of lag times between the two flood drivers. In our revised manuscript, we expand Section 5.4 in lines 489-494 by adding:

Different locations of rain gauges may introduce timing lags and lead to uncertainties in defining "simultaneous" extremes. However, the timing differences of NTR across New York Harbor, e.g. in Jamaica Bay, off Manhattan, or in Newark Bay, are at most 30 minutes based on the shallow water wave travel time (similar to tide) from offshore to reach these locations which have pathways with distances of at most 20 km. For single-gauge analysis of rainfallsurge timing, these location differences may help explain different rank correlations. However, for the joint probability analysis and lag time histograms we are using a spatial average rainfall, which captures regions surrounding the tide gauges well and should introduce very little timing difference.

2. Line 46: "and frameworks". Elaborate on what frameworks you refer to here, it is current statement is quite vague.

We appreciate the comment regarding the lack of clarity. In the revised version, we now specify that by "frameworks" we refer to established multivariate statistical and probabilistic modeling frameworks—such as Copula-based models and joint return period analyses—that have been widely used to assess the dependency between compound flood drivers. This clarification is added in line 47 to enhance the reader's understanding.

3. "Metro-scale rainfall". The reason for estimating average over that scale and its incorporation in the overall analysis is not clearly explained.

We agree that the role of metro-scale rainfall requires further explanation. In our revised manuscript, we clarify in lines 147-149 that calculating a spatial average over rain gauges within a 30-km radius helps smooth out the localized variability in rainfall, giving a perspective more reflecting an integrated hourly (or longer) effect on flooding. Averaging

gages over such a spatial scale is a common approach for compound flood studies. As already noted in the text, we analyzed both the sewershed scale rain data (single-gauge) and the metro-scale rainfall in our analyses, for different perspectives.

4.L150: "we eliminate peaks that occur within 5-day windows". I assume that you mean that only the max peak within a 5-day window was retained(?). Please clarify. 5 days is admittedly a long duration for small scale pluvial flood events.

We confirm that our methodology retains only the maximum peak within any 5-day window to ensure event independence. This 5-day window was chosen to account for the typical maximum duration of cyclonic storm events. In lines 156-157, we clarify this point in the revised manuscript and discuss the rationale behind choosing a 5-day interval.

5. What is the total (and per class) sample size of rainfall and NTR peaks? The exact numbers should be reported for the readers to appreciate the sample size involved in this analysis.

We agree that providing explicit numbers will help readers better appreciate the robustness of our analysis. In the revised manuscript, we include a table that summarizes the total number of rainfall and NTR peaks extracted from the 75-year record, along with a breakdown by storm type (TC, ETC, and non-cyclonic/convective events).

6. Figure 3 is not visually appealing (just the opinion of this reviewer), I wonder if you could improve how you convey the information in this figure.

We appreciate the reviewer's aesthetic feedback. We revise Figure 3 by exploring alternative color palettes (for example, those recommended by ColorBrewer) and refining the graphical layout to enhance clarity and readability. Our goal is to improve the figure's ability to convey key information regarding the annual frequency of compound events across different storm types.

7. Figure 6: Have you accounted for the bin width when you calculated the values? If not the y-axis should be labeled frequency instead of probability density.

We agree with the reviewer that the y-axis for Figure 6 should simply be "Frequency (1/year)" and we change the figure.

8. Finally, considering that the compound flood hazard (i.e. total flood depth resulting from rainfall and surge) is not explicitly considered and thus conclusions on the influence of the storm types on the hazard cannot be directly derived without coupled simulations (as the author acknowledge), I would recommend modifying the title to "Influence of Storm Type on Compound Flood Drivers" or something along those lines.

We appreciate the suggestion. In recognition that our analysis primarily addresses the influence of storm type on the flood drivers (rainfall and surge) rather than on the combined flood depth, we agree that a revised title would more accurately reflect our study's scope. Accordingly, we agree with changing the title to "Influence of Storm Type on Compound Flood Drivers." We believe this modification will better set the expectations regarding the focus of our work.